# Urban Development Index (UDI): A Comparison between the City of Rio de Janeiro and Four Other Global Cities

**Rafael Molinaro [1], Mohammad K. Najjar [2]** **, Ahmed W. A. Hammad [3], Assed Haddad [4]** and **Elaine Vazquez [5],***

1   Departamento de Construção Civil, Universidade Federal do Rio de Janeiro, Rio de Janeiro 21941-909, Brazil; rafael_molinaro@poli.ufrj.br
2   Centro Universitário Gama e Souza (UNIGAMA), Rio de Janeiro 22621-090, Brazil; mnajjar@poli.ufrj.br
3   Faculty of Built Environment, UNSW Sydney, Sydney 2052, Australia; a.hammad@unsw.edu.au
4   Programa de Engenharia Ambiental, Universidade Federal do Rio de Janeiro, Rio de Janeiro 21941-909, Brazil; assed@poli.ufrj.br
5   Programa de Engenharia Urbana, Universidade Federal do Rio de Janeiro, Rio de Janeiro 21941-909, Brazil
*   Correspondence: elaine@poli.ufrj.br

**Abstract:** One of the methods to assess the urban development of a city is to allocate indicators that quantify its efficiency in performing various functions, such as urban mobility, security, sustainability, and economy, among others. The motivation of this work is the fact that the city of Rio de Janeiro, although widely known and admired around the world for its natural beauty, has a wide negative notoriety regarding its urban functionality. There is a critical need for investment in the city's infrastructure in order to improve the quality of life of its population. The novelty of this work is in proposing an index that quantifies the urban functionality of the city of Rio de Janeiro and that represents urban development. The research focuses on optimizing investments in infrastructure and hence increasing the urban performance offered by the city of Rio de Janeiro. In the proposed methodology for modeling the Urban Development Index (UDI), this work presents the model structure made from a knowledge-based urban development assessment model (KBUD/AM), the determination of the indicators, the selection of the cities, the data collection from primary and secondary sources and the use of statistical techniques for data normalization and scaling. The research aims to quantify, compare and evaluate the level of urban development of Rio de Janeiro, performing benchmarking with other four global cities (Stockholm, Shanghai, Boston, and Cape Town). Cities are ranked for their UDI to make the comparison more straightforward. Based on the results, Rio de Janeiro ranks second to last among the five cities studied, with an UDI of 0.395, only slightly better than Cape Town. The results confirm that the city of Rio de Janeiro has several deficiencies, especially in the education, safety and health sectors, and is very far from most of the other developed cities. The city of Rio de Janeiro should promote investments in research and development. Finally, this work confirms that Rio de Janeiro must tackle security problems as a matter of priority.

**Keywords:** Urban Development Index; knowledge-based urban development (KBUD); comparative analysis; Rio de Janeiro; Brazil

## 1. Introduction

Cities can be characterized as a set of interconnected systems comprised of transportation and energy networks, waste collection depots, sewage treatment plant, paving and urban buildings. Such

elements make up the urban infrastructure and provide a high quality of life for its inhabitants [1–4]. The quality of life in an urban region is influenced by several elements, such as environmental quality, safety, quality and service delivery, government and others [5]. To assess the performance of a city (i.e., the level of functioning), the Urban Development Index (UDI) can be utilized, which provides a measure of the level of sustainable development in a city [6]. In this context, there is a high necessity to determine patterns and predictions that allow the evaluation of the UDI, a process that involves the amalgamation of several indicators to guide better policy decisions and resource allocations for the development of a society [7].

The city of Rio de Janeiro is the second largest city in Brazil in terms of population, with more than 12 million residing in it (6.0% of the national population) [8]. The city is recognized as the Brazilian cultural capital and has recently received the title of First World Capital of Architecture by the United Nations Educational and Cultural Organization (UNESCO) [9]. Unfortunately, the city is also linked to the huge presence of slums (urban conglomerates of the low-income population, usually illegally occupied) that show the great social disparity experienced by the entire Brazilian population, along with levels of insecurity to which they are subjected [10]. In 2010, Rio de Janeiro was presented in a favorable context on the international scene, with the victory in its applications to host the 2014 World Cup and the 2016 Olympics [11], in addition to other major events such as Rio +20 in 2012, World Youth Day 2013 and the Confederations Cup 2013. Unfortunately, at the 2014 World Cup and the 2016 Olympics, the "wonderful city" was taken to mean insecurity, poor infrastructure and poor quality of life, suffering criticism from the international press [12,13]. Furthermore, the city is facing critical urban problems with the advent of more expensive, more exclusionary, more fragmented and more privatized city districts [14,15]. Other problems such as the high level of homicide, thefts and the dominance of criminal groups over some territories are pointed out as negatively impacting public safety [16]. Urban mobility issues (i.e., how to make a person transition from the car to public transport) [17], and the lack of public policies for housing and sanitation generate a disorderly and irregular sewage disposal [18]. According to the United Nations, Brazil occupies the 79th ranking out of 189 countries in terms of the highest Human Development Index (HDI), behind countries like Venezuela, Sri Lanka and Cuba [19]. Meanwhile, the municipality of Rio de Janeiro occupied the 45th rank amongst the Brazilian municipalities with the highest HDI [20].

The unstructured urban environment of Rio de Janeiro requires the application of more investments in the infrastructure of the city to improve the quality of life of its population [21,22]. In addition, a reference point is required to assess the actual condition expressed by the city [6]. At this level of the analysis, performing benchmarking against other cities around the world could facilitate understanding and exposing the underlying issues with the urban performance of Rio de Janeiro [23].

*Justification for the Study*

Quality of life in cities is a multidimensional subject and, in this respect, its evaluation is a challenge [24]. In the recent literature, a large number of studies focused on the quality of life in cities [25–27]. The environmental index is strongly connected with quality of life [24,28,29]. Other indices also affect the quality of life in cities, such as governance, economy, mobility, safety, quality and service provision [30]. Good urban performance is directly related to effective and balanced joint development between the social, environmental and economic sectors of sustainable development [31–34].

Studies in the literature have highlighted that many factors can directly or indirectly affect sustainable urban development levels, indicating that the potential index system scale is large [35]. This research aims to quantify, compare and evaluate the level of urban development in Rio de Janeiro in relation to four other chosen global cities (Stockholm, Shanghai, Boston, and Cape Town). The novelty of this work is in determining an index, denominated as the UDI, that quantifies the urban functionality of the city, and that represents urban development. The research focuses on optimizing the investments in the infrastructure and increasing the urban performance offered by the city of

Rio de Janeiro. Data-based modeling was applied from the knowledge-based urban development (KBUD) methodology, in order to obtain a UDI for Rio and the other selected cities. As a scientific contribution, this study will make it possible to observe the particularities of the urban functioning of Rio de Janeiro compared to the other cities studied, as well as identifying the possibilities of urban evolution.

## 2. Materials and Methods—Urban Development Index (UDI)

In this section, the definition of the model structure to be used in this study must be defined as a prior step for the modeling of the UDI. Once the structure is defined and based on Carrillo et al., 2014 [36], the following steps are taken: (i) determination of the indicators that will provide the data for the formulation of the UDI; (ii) selection of cities; (iii) data collection from primary and secondary sources; (iv) use of statistical techniques for data normalization and scaling. The following items describe in detail the definition of model structure and the next four steps of the methodology.

### 2.1. Definition of the Model Structure

There are several models for quantifying urban development; however, many of these models consider only a category of analysis or a limited number of them, being somewhat superficial to be applied to the elaboration of a representative index [37]. In addition, one should choose a model that can be used for all cities analyzed, regardless of their development pattern or geographic location. In these terms, HDI is the most widespread and well-known index when comparing countries, cities or localities in general [38]. This index was first created in 1990 and then modified in 2010, taking into consideration three components: (i) life expectancy, (ii) education and (iii) gross domestic product (GPD) per capita [39]. The index has several flaws, becoming redundant and generally presenting the same result obtained with GDP per capita, as well as evaluating only three components that do not faithfully reflect what is intended to be measured [40].

In the literature, several studies presented some of the most popular indices in the scientific academy and international bodies such as the City Development Index (CDI), the Global Cities Index (GCI), the Global Power City Index (GPCI) and the Global City Competitiveness Index (GCCI). [41,42]. Some of these indices use models that encompass few or only one category of analysis and are too low to analyze the urban performance of a city as a whole [37]. In these terms, this paper presents a proposal for modeling a new index, absorbing the most relevant indicators and parameters to target the scope of satisfactorily representing urban development from Rio de Janeiro. Therefore, the starting point is to adopt the knowledge-based urban development assessment model (KBUD) [43–48], as it is an indicator-based evaluation model that aims to measure the level of urban development, and can undergo changes that meet the desired analysis [49].

The KBUD methodology was applied based on the notion that, with the development of society and the means of production, cities have goals to attract and retain knowledge resources, technologies, patents, and productivity. In other words, the old model of the search for natural and physical means (commodities, ores, raw materials) has been replaced with one that is focused on human capital, services, and products with an informational origin of knowledge [49,50]. This model has been used in cities that want to improve their infrastructure, attract more investments and provide prosperity and high quality of life to their inhabitants. [51,52] Initially, KBUD was implemented as a form of urban development in small cities only. Then, it was developed and applied with large global knowledge bases such as Austin, Barcelona, Helsinki, Manchester, Melbourne and Singapore. Recently, there has been an increase in the application of KBUD in emerging cities such as Beijing, Dubai, Istanbul, Kuala Lumpur, Monterrey and Shenzhen [53–55].

The KBUD methodology consists of an index composed of 32 singular indicators that form eight groups and four categories. The 32 unique indicators measure features such as the analytical capacity, comparability, geographic coverage, availability and relevance of data [50,52,54,55]. When collecting the associated data, it was necessary to perform a normalization process of the singular indicators,

since most data have different units and orders of magnitude and it was impracticable to apply a simple arithmetic operation. Taking into consideration the already normalized indicators, an equal weight was adopted for the eight groups of indicators, which were macroeconomics (ME), innovation and technology (IT), human and social capital (HSC), diversity and independence (DI), sustainable urban development (SUD), quality of life and place (QLP), governance and planning (GP), and leadership and support (LS) [6]. In these terms, the ME group quantitatively represents the economy of the examined city (i.e., the economic size of the city and its inhabitants worldwide) [56,57]. The IT group illustrates the effort of the city to update and modernize its economic production [56,57]. The HSC group seeks to measure the individual and collective development of the population [58,59]. The DI group describes the characteristics of the populations of cities and the level of social imposition applied by their governments [56,57]. The SUD group characterizes the attempt to create urban spaces that remain functional and viable over time [56,57]. The QLP group assesses the urban conditions that allow for good living and housing in the cities [56,57]. The GP group addresses the macro issues of governance, attractiveness and strategic planning of each city [56,57]. The LS group analyzes the mechanisms used by the national government of each city to ensure transparency and good economic and social results [56,57]

Similarly, equal weight was adopted for each group with each two forming the four categories of mid-level indicators that refer to the four development pillars of KBUD, namely economy, society, environment and governance [6,58,59]. Finally, equal weight is adopted for each category, which together formed the overall high-level composite indicator [6].

## 2.2. Indicators Determination to Formulate the UDI

In line with the KBUD approach presented in the previous subsection, the proposed model of this study consisted of an index composed of the same 32 singular indicators. Singular indicators were normalized and were then divided into eight groups, namely (ME), (IT), (HSC), (DI), (SUD), (QLP), (GP), and (LS) [56].

The presented eight groups were then paired in order to determine the four main categories of analysis that provided one indicator and which referred to the three pillars of sustainable development (economy, society, and environment), as well as the fourth pillar (government). Overall, the indicators were economy and technology, society and culture, urban and environmental, and government [60]. Each field of analysis provided an independent indicator: economic indicator (EI), social indicator (SI), environmental indicator (EnI) and government indicator (GI) [60]. The EI encompasses the economic and technological fields. This indicator represents quantitatively and qualitatively the capacity of each city to generate economy and innovation, leveraging its financial and technological power [60]. The SI represents the social and cultural fields, and refers to the characteristics of the population of each city, taking into consideration its intellectual and cultural productive capacity and its diversified and sustainable development [60]. The EnI characterizes the urban structuring and organization, involving the environmental and housing sustainability of cities [60]. The GI indicates the level of effectiveness and transparency of government institutions in securing the means for a well-functioning city [60]. Finally, a single global index was composed by adopting an equal weight for each of the four categories, which resulted in the UDI [60].

## 2.3. City Selection

The use of the term benchmarking may be beneficial for a better understanding of the index, as it provides a reference point for the interpretation of measured results [23]. More appropriately, the term city benchmarking was used, which was defined as a "system of socioeconomic indicators that constitute a control panel in the diagnostic phase of the urban strategic planning process" [61]. Considering such comparative analysis, it was possible to identify the key elements, peculiarities, and deficits that will help define the future strategy to be implemented in urban development [62]. Thus, a city benchmarking with five other cities was performed in order to obtain a more accurate analysis of

the values found for the indicators and the UDI of the city of Rio de Janeiro, aiming at obtaining the city position in relation to the others.

*2.4. Collection of Data*

The next step after selecting cities was to collect data from the 32 indicators using primary and secondary sources. At this level of the analysis, Table 1 presents the eight indicator groups. In addition, Table 1 illustrates the four main categories of analysis that provided independent indicators such as (EI), (SI), (EnI) and (GI), as well as their 32 unique indicators. Table 2 provides an explanatory description of each of the 32 singular indicators and bibliographic references.

**Table 1.** Description of the eight indicator groups, four analysis categories, and their 32 singular indicators.

| Analysis Categories | Indicator Groups | Singular Indicators |
|---|---|---|
| economic indicator (EI) | macroeconomics (ME) | GDP<br>International companies<br>Foreign direct investment<br>Urban competitiveness |
| | innovation and technology (IT) | Innovation<br>Research and Development<br>Smart city<br>Patents |
| social indicator (SI) | human and social capital (HSC) | Education<br>Universities<br>Health<br>Connectivity |
| | diversity and independence (DI) | Immigration<br>Freedom<br>Socioeconomic Dependence<br>Unemployment |
| environmental indicator (EnI) | sustainable urban development (SUD) | Environmental impact<br>Sustainability<br>Urban density<br>Urban mobility |
| | quality of life and place (QLP) | Quality of life<br>Cost of living<br>Residency<br>Safety |
| government indicator (GI) | governance and planning (GP) | Government Effectiveness<br>Electronic Governance<br>Tourism<br>Urban planning and resilience |
| | leadership and support (LS) | Corruption<br>Taxes<br>Inflation<br>Social equality |

**Table 2.** Description of the 32 singular indicators and their bibliographic references.

| Singular Indicators | Description and Bibliographic Reference |
|---|---|
| GDP | Gross domestic product (GDP) per capita in purchasing power parities in USD [63] |
| International companies | Number of headquarters of the world's 500 largest companies by market value [64] |
| Foreign direct investment | List of international participation in foreign direct investment [65] |
| Urban competitiveness | Global Urban Competitiveness Index Score [66] |
| Innovation | Global Innovation Index Score [67] |
| Research and Development | Share of GDP for research and development [68] |
| Smart city | Smart cities global index score [69] |
| Patents | Patent Cooperation Treaty patent applications per million inhabitants [70] |
| Education | The ratio of public spending on education to GDP [71] |
| Universities | University ranking best placed international ranking [72] |
| Health | Value in International Health Index [73] |
| Connectivity | National inhabitants with fixed broadband access per 100 inhabitants [74] |
| Immigration | International Ranking Score in the Immigration Tolerance Category [75] |
| Freedom | International Personal Freedom Index Score [76] |
| Socioeconomic Dependence | The ratio between the elderly and the economically productive population [77] |
| Unemployment | Unemployment Rate [78] |
| Environmental impact | $CO_2$ emissions in metric tons per capita [79] |
| Sustainability | Global ranking of sustainable cities [80] |
| Urban density | Population density in inhabitants per $km^2$ [81] |
| Urban mobility | Global Sustainable Urban Mobility Ranking [82] |
| Quality of life | International Ranking in Quality of Life [83] |
| Cost of living | International Ranking on Cost of Living [84] |
| Residency | Accessibility to the housing by the international index [85] |
| Safety | International Index on Personal Safety [86] |
| Government Effectiveness | Government Effectiveness [87] |
| Electronic Governance | International Electronic Government Index Score [88] |
| Tourism | Number of international visitors per year [89] |
| Urban planning and resilience | Ranking in International Index for Urban Planning and Resilience [90] |
| Corruption | International Corruption Index Score [91] |
| Taxes | Percentage of a tax burden on GDP [92] |
| Inflation | Inflation Rate [93] |
| Social equality | Income inequality level in gini coefficient [94] |

*2.5. Data Normalization*

Statistical normalization was performed to scale values in relation to the maximum and minimum values available in the research dataset. Thus, it was expected to obtain a numerical value that represented the performance of cities in terms of the full sampling available. A value of (1) represented that the city occupies the best position, or holds the best result for that indicator worldwide, or at least among all the cities analyzed in the sources. Similarly, a value of (0) corresponded to the worst possible performance for that singular indicator. The data normalization was conducted in accordance with Equation (1), as follows:

$$I_{norm} = \frac{I_{gross} - I_{min}}{I_{max} - I_{min}} \tag{1}$$

where:

$I_{norm}$—normalized value of the singular indicator;

$I_{gross}$—gross value;

$I_{min}$—minimum value among all present in the consulted source;

$I_{max}$—maximum value among all present in the consulted source.

The direction of some indicators was evaluated, and the normalized value of (1) was subtracted for the negative or ranking indicators. Twelve indicators were adjusted, namely socioeconomic dependence, unemployment, environmental impact, sustainability, urban density, urban mobility, quality of life, cost of living, urban planning and resilience, taxes, inflation, and social equality. For

the cases of indicators of negative character or denoting rankings, normalization was conducted via Equation (2), as follows:

$$I_{norm} = 1 - \frac{I_{gross} - I_{min}}{I_{max} - I_{min}} \tag{2}$$

where:

$I_{norm}$—normalized value of the singular indicator;
$I_{gross}$—gross value;
$I_{min}$—minimum value among all present in the consulted source;
$I_{max}$—maximum value among all present in the consulted source.

After normalizing the 32 indicators, equal weight was adopted for all indicators that formed the lower-level eight groups of indicators already cited, from an arithmetic average of four indicators. Similarly, an equal weight was adopted for each group which formed the mid-level four categories of indicators already mentioned. Finally, equal weight was adopted for each category, which together formed the higher-level global composite indicator, giving the UDI measure, as presented in Equation (3).

$$UDI = \sum_{i=1}^{n} \frac{CAT_i}{n} \tag{3}$$

where:

*UDI*—value of the Urban Development Index;
$CAT_i$—categories that make up the UDI;

## 3. Results

Benchmarking Rio De Janeiro against five other cities was performed in order to obtain a more accurate analysis of the values found for the UDI. The first criteria for choosing cities was their requirement to be global cities [95], and their geographical position. This work proposes selecting cities with very distinct locations, which in themselves have unique characteristics, cultures, and particularities, as well as different geographical positions. Two cities that have a very high urban development in comparison to Rio de Janeiro were considered, with the cities being amongst ones with the highest urban performance in the world, namely Stockholm in Sweden and Boston in the United States. These cities will be referred to as the "model cities". Two cities that have development characteristics comparable to Rio de Janeiro were filtered in a preliminary analysis. They are important cities of emerging countries and they include Shanghai in China and Cape Town in South Africa [96]; these cities are referred to as "comparative cities". Considering the criteria adopted, the UDI of Rio de Janeiro was expected to be below the model cities and to rank among the comparative cities. Figure 1 illustrates the countries of origin of the cities studied and locates them geographically.

Defining the structure of the adopted model, as well as the selection of the examined five cities could facilitate the collection of data for 32 indicators in the five examined cities. The gross values were then obtained, which were normalized according to Equations (1) and (2) to homogenize the quantities and to scale the values obtained for the examined cities. Then, the indicators were joined to form the lower-level eight groups and four mid-level categories of analysis, in addition to the high-level composite UDI, which measures the various fields of urban functionality considered. The results of the analysis are presented in absolute values, rankings, and radar charts, facilitating the visualization and interpretation of the information.

Some characteristics and primary information of the analyzed cities are presented in Table 3. The cities were ranked based on their national rankings in the latest HDI update [19].

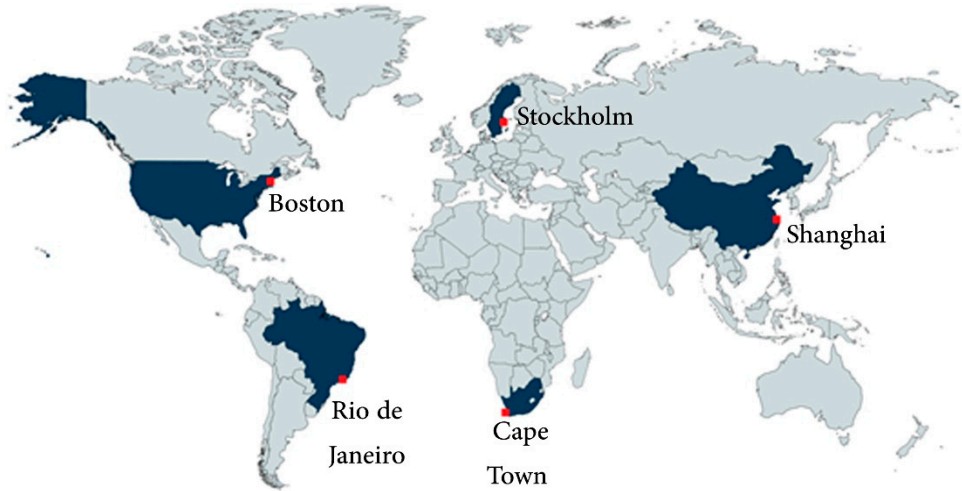

**Figure 1.** Cities considered in the analysis.

**Table 3.** Primary characteristics of the studied cities.

| City | Country | Population | Area (km$^2$) | HDI Ranking |
|---|---|---|---|---|
| Rio de Janeiro | Brazil | 11,990,000 | 1.917 | 79° |
| Stockholm | Sweden | 1,565,000 | 414 | 7° |
| Boston | United States | 7,315,000 | 9.189 | 13° |
| Shanghai | China | 24,115,000 | 4.015 | 86° |
| Cape Town | South Africa | 3,980,000 | 816 | 113° |

*3.1. Presentation of Absolute Values of 32 Singular Indicators*

The practical description of the 32 singular indicators in the five examined cities is presented in Table 4, considering the maximum and minimum value for each indicator. It is noteworthy to point out that all data were obtained from the references indicated in Table 2. However, a clear difference can be observed between the values of model cities and the values of comparative cities. The former generally have better values for the indicators studied. For the maximum and minimum values, it is important to declare that outliers—atypical values very discrepant in relation to the others - were disregarded so as not to impair the normalization process.

**Table 4.** Absolute values of the 32 maximum and minimum indicators.

| Indicators/Cities | Rio de Janeiro | Stockholm | Boston | Shanghai | Cape Town | Max. | Min. |
|---|---|---|---|---|---|---|---|
| GDP | 14,176.0 | 56,250.0 | 76,204.0 | 24,065.0 | 14,086.0 | 93,849.0 | 4036.0 |
| International companies | 2 | 1 | 2 | 7 | 0 | 20 | 0 |
| Foreign direct investment | 1.61% | 1.31% | 14.18% | 4.16% | 0.46% | 15.60% | 0.00% |
| Urban competitiveness | 0.42 | 0.79 | 0.81 | 0.84 | 0.33 | 1.00 | 0.00 |
| Innovation | 41 | 48 | 53 | 47 | 37 | 56 | 16 |
| Research and Development | 1.17% | 3.16% | 2.74% | 2.11% | 0.73% | 4.30% | 0.07% |
| Smart city | 3.94 | 6.95 | 6.81 | 4.6 | 3.46 | 7.24 | 2.76 |
| Patents | 3.45 | 320.11 | 173.14 | 15.2 | 6.26 | 335.16 | 0 |
| Education | 5.90% | 7.70% | 4.90% | 4.00% | 6.00% | 14.60% | 0.60% |
| Universities | 30.30 | 59.30 | 100.00 | 77.60 | 43.90 | 100.00 | 23.50 |
| Health | 43.33 | 66.01 | 76.46 | 60.45 | 72.13 | 85.45 | 37.45 |
| Connectivity | 13.70 | 37.70 | 33.85 | 26.86 | 2.99 | 50.25 | 0.00 |
| Immigration | 7.20 | 8.70 | 8.50 | 2.80 | 6.30 | 10.00 | 0.00 |
| Freedom | 69.39 | 88.07 | 78.30 | 37.88 | 77.77 | 92.07 | 25.19 |
| Socioeconomic Dependence | 43.80 | 58.50 | 51.20 | 37.70 | 52.50 | 17.40 | 111.60 |
| Unemployment | 13.10% | 6.30% | 4.10% | 3.90% | 27.50% | 0.30% | 50.00% |
| Environmental impact | 5.03 | 5.29 | 19.9 | 8.49 | 9.49 | 0.32 | 54.41 |
| Sustainability | 84 | 2 | 22 | 76 | 97 | 1 | 100 |
| Urban density | 6300 | 3700 | 800 | 6000 | 4900 | 500 | 47,400 |
| Urban mobility | 63 | 9 | 46 | 27 | 66 | 1 | 100 |
| Quality of life | 118 | 23 | 35 | 103 | 94 | 1 | 231 |
| Cost of living | 99 | 89 | 70 | 7 | 170 | 1 | 209 |
| Residency | 0.34 | 1.14 | 1.53 | 0.29 | 0.9 | 9.87 | 0.03 |
| Safety | 22.22 | 52.61 | 67.68 | 46.33 | 30.03 | 88.26 | 14.82 |
| Government Effectiveness | 41.80 | 96.20 | 92.80 | 68.30 | 65.40 | 100.00 | 0.00 |
| Electronic Governance | 0.73 | 0.89 | 0.88 | 0.68 | 0.66 | 0.92 | 0.06 |
| Tourism | 1370 | 2080 | 1740 | 6120 | 1370 | 21,470 | 710 |
| Urban planning and resilience | 126 | 16 | 21 | 57 | 143 | 1 | 165 |
| Corruption | 37 | 84 | 75 | 41 | 43 | 89 | 9 |
| Taxes | 34.40% | 49.80% | 26.00% | 20.10% | 26.90% | 1.40% | 64.07% |
| Inflation | 4.17% | 1.60% | 2.10% | 1.80% | 5.40% | −0.90% | 41.50% |
| Social equality | 48.70% | 24.90% | 47.00% | 42.20% | 62.50% | 23.70% | 63.20% |

*3.2. Presentation of the Normalized Values of the 32 Singular Indicators*

The normalization of the values in Table 4 is presented in Table 5, according to Equations (1) and (2).

Attention should be given to the character of the discussed indicators, whose lower absolute numbers represent a better performance (rankings or negative indicators); these should have their normalization adjusted so that a lower gross value sets a higher normalized value. Due to the applied normalization, considering the maximum and minimum values among all available sources, and not only among the analyzed cities, there are also parameters whose values for the five studied cities are totally low or high. This means that for that parameter, all five cities are far from extreme values. This setting is easily observed in the parameter "World Companies", where the highest normalized value is only (0.350). This is due to the fact that the major world cities—New York, Los Angeles, London, Paris, Tokyo and Beijing—which have the largest headquarters of these companies, are not present in the analysis [95].

Similarly, in the "housing" category, all cities have low values; that is, all cities are considered expensive to live in, a situation already expected because they are metropolises with high economic prospects. In contrast, the five cities obtained good values in the parameters "Urban density" and "Electronic governance". The first is due to the fact that no city analyzed is part of South Asia, the region with the worst urban density [82]. The second is related to the fact that all cities are part of developed or emerging countries, while the source researched has data from various countries of the world, including highly underdeveloped nations such as Central African countries [88].

**Table 5.** Normalized values of the 32 indicators.

| Indicators/Cities | Rio de Janeiro | Stockholm | Boston | Shanghai | Cape Town |
|---|---|---|---|---|---|
| GDP | 0.113 | 0.581 | 0.804 | 0.223 | 0.112 |
| International companies | 0.100 | 0.050 | 0.100 | 0.350 | 0.000 |
| Foreign direct investment | 0.103 | 0.084 | 0.909 | 0.267 | 0.029 |
| Urban competitiveness | 0.424 | 0.786 | 0.812 | 0.837 | 0.328 |
| Innovation | 0.625 | 0.800 | 0.925 | 0.775 | 0.525 |
| Research and Development | 0.260 | 0.731 | 0.632 | 0.482 | 0.157 |
| Smart city | 0.263 | 0.935 | 0.904 | 0.411 | 0.156 |
| Patents | 0.010 | 0.955 | 0.517 | 0.045 | 0.019 |
| Education | 0.379 | 0.507 | 0.307 | 0.243 | 0.386 |
| Universities | 0.089 | 0.468 | 1.000 | 0.707 | 0.267 |
| Health | 0.123 | 0.595 | 0.813 | 0.479 | 0.723 |
| Connectivity | 0.273 | 0.750 | 0.674 | 0.534 | 0.060 |
| Immigration | 0.720 | 0.870 | 0.850 | 0.280 | 0.630 |
| Freedom | 0.661 | 0.940 | 0.794 | 0.190 | 0.786 |
| Socioeconomic Dependence | 0.720 | 0.564 | 0.641 | 0.785 | 0.627 |
| Unemployment | 0.742 | 0.879 | 0.924 | 0.928 | 0.453 |
| Environmental impact | 0.913 | 0.908 | 0.638 | 0.849 | 0.830 |
| Sustainability | 0.162 | 0.990 | 0.788 | 0.242 | 0.030 |
| Urban density | 0.876 | 0.932 | 0.994 | 0.883 | 0.906 |
| Urban mobility | 0.374 | 0.919 | 0.545 | 0.737 | 0.343 |
| Quality of life | 0.491 | 0.904 | 0.852 | 0.557 | 0.596 |
| Cost of living | 0.529 | 0.577 | 0.668 | 0.971 | 0.188 |
| Residency | 0.032 | 0.113 | 0.152 | 0.026 | 0.088 |
| Safety | 0.101 | 0.515 | 0.720 | 0.429 | 0.207 |
| Government Effectiveness | 0.418 | 0.962 | 0.928 | 0.683 | 0.654 |
| Electronic Governance | 0.788 | 0.969 | 0.956 | 0.728 | 0.705 |
| Tourism | 0.032 | 0.066 | 0.050 | 0.261 | 0.032 |
| Urban planning and resilience | 0.238 | 0.909 | 0.878 | 0.659 | 0.134 |
| Corruption | 0.350 | 0.938 | 0.825 | 0.400 | 0.425 |
| Taxes | 0.473 | 0.228 | 0.607 | 0.702 | 0.593 |
| Inflation | 0.899 | 0.934 | 0.929 | 0.941 | 0.854 |
| Social equality | 0.367 | 0.970 | 0.410 | 0.532 | 0.018 |

## 3.3. Results of the Eight Indicators Groups

Starting from the normalized values, an arithmetic average was made for every four indicators, forming the eight analysis groups; ME, IT, HSC, DI, SUD, QLP, GP, and LS. These values are presented in Table 6.

**Table 6.** Indicators of the eight Groups.

| Indicators/Cities | Rio de Janeiro | Stockholm | Boston | Shanghai | Cape Town |
|---|---|---|---|---|---|
| ME | 0.185 | 0.375 | 0.656 | 0.419 | 0.117 |
| IT | 0.290 | 0.855 | 0.744 | 0.428 | 0.214 |
| HSC | 0.216 | 0.580 | 0.698 | 0.491 | 0.359 |
| DI | 0.711 | 0.813 | 0.802 | 0.545 | 0.624 |
| SUD | 0.581 | 0.937 | 0.741 | 0.678 | 0.528 |
| QLP | 0.288 | 0.527 | 0.598 | 0.496 | 0.270 |
| GP | 0.369 | 0.726 | 0.703 | 0.582 | 0.381 |
| LS | 0.522 | 0.767 | 0.693 | 0.644 | 0.472 |

The output results of Table 6 can be summarized as follows:

- For the ME group, the scarcity of headquarters of world companies present in the studied cities compared to the main world cities decreases the group indicator values for all. Boston ends up

having the best result largely because of the large economy of the USA [97], which guarantees high values in foreign direct investment and per capita GDP indicators.

- For the IT group, Stockholm presents itself as the best performing city, with high indicators in all four parameters. It is essential that cities that have not performed well in macroeconomic measures, and therefore do not yet have a solid economic base, try to improve their economic development by investing in innovation and technology.
- For the HSC group, the group indicator is broadly aligned with the HDI [19], with the exception of Boston, which has high performances due to the university indicator.
- For the DI group, Stockholm is the most diverse and independent city among the five cities.
- For the SUD group, Stockholm presents the best balance between sustainable urban land use and environmental impacts. Even on many islands, Stockholm has an organized and connected urban network and a clean energy matrix [98].
- For the QLP group, the city of Boston has the best result, which also reflects a good economy [97].
- For the GP group, the Stockholm group stands out for its good strategic planning [98].
- For the LS group, once again the excellent performance of Stockholm is highlighted, which despite not being the leading city, presents excellent values considering its poor performance due to its high taxes, characteristic of the welfare state, and Scandinavian social welfare [99].

### 3.4. Results of the four Indicators Category

The eight groups were then joined in pairs, determining the four main categories of analysis: economic-technological, socio-cultural, urban-environmental and governmental in order to obtain one indicator for each: EI, SI, ENI, and GI, respectively, as presented in Table 7.

**Table 7.** Indicators of the four Categories.

| Indicators/Cities | Rio de Janeiro | Stockholm | Boston | Shanghai | Cape Town |
|:---:|:---:|:---:|:---:|:---:|:---:|
| EI | 0.237 | 0.615 | 0.700 | 0.424 | 0.166 |
| SI | 0.463 | 0.697 | 0.750 | 0.518 | 0.491 |
| EnI | 0.435 | 0.732 | 0.670 | 0.587 | 0.399 |
| GI | 0.446 | 0.747 | 0.698 | 0.613 | 0.427 |

The output results of Table 7 can be summarized as follows:

- For the EI measure, Boston achieves the best result among the five cities, followed by Stockholm, driven by the major USA national economies [97], and Swedish technological and innovative pioneerism [70], respectively.
- For the SI measure, it highlights an unexpected low for Stockholm city, where the presence of the Scandinavian welfare state would indicate good results for this category [98].
- For the EnI measure, it characterizes the urban structuring and organization involving the environmental and housing sustainability of cities. For this category, the best performing city was Stockholm. The worst places were Rio de Janeiro and Cape Town, respectively.
- For the GI measure, it indicates the level of effectiveness and transparency of government institutions in securing the means for a well-functioning city. Because of this, model cities have high values, with an average of 0.722, while comparative ones naturally have lower values, with an average of 0.520.

### 3.5. Determination of Urban Development Index (UDI)

The UDI for each examined city is the result of grouping the four categories presented in Table 7. Hence, the final values obtained for the composite indices are shown in Table 8, which includes three rows. The ranking positions in the HDI of these cities and the final ranking in relation to the HDI and UDI are presented, in order to assist the observation of how the cities are positioned among themselves.

**Table 8.** Urban Development Index (UDI).

| Indicators/Cities | Rio de Janeiro | Stockholm | Boston | Shanghai | Cape Town |
|---|---|---|---|---|---|
| Indicator UDI | 0.395 | 0.698 | 0.705 | 0.535 | 0.371 |
| Final Ranking UDI | 4 | 2 | 1 | 3 | 5 |
| Ranking Position HDI | 79° | 7° | 13° | 86° | 113° |
| Final Ranking HDI | 3 | 1 | 2 | 4 | 5 |

The first conclusion is the non-confirmation of the relationship between the UDI and the HDI. The values do not follow the same linearity as the HDI. Stockholm, for example, has a worse performance in comparison to Boston for the UDI, while it has the highest HDI compared to the same city, ranking 7th worldwide [19]. Additionally, Shanghai performs well below the model cities, but well above Cape Town and Rio de Janeiro.

Rio de Janeiro, the reference city of the work, ranks as the second to last among the five examined cities, obtaining a better income only than the South African city of Cape Town. It should also be pointed out that the values do not represent a scale where a 1000 UDI denotes the city with the largest urban development in the world and one with a value of 0.000, the worst. Such conditions are impossible, since for a city to reach the UDI value of 1000 (0.000), it should be the best/worst of all sources, for the 32 indicators.

## 4. Discussion

Having all the results for the singular indicators, groups, categories, and UDI facilitates interpreting and discussing the indicators for the five cities and determining what each city should look for to improve its urban performance. As this work focuses on the city of Rio de Janeiro, the results are shown in radar graphs for an easy interpretation and comparison of the outputs. Hence, the two model cities will be presented on the right of the graphs, the two comparative cities will be presented on the left of the graphs, and Rio de Janeiro is presented in the center of the graphs.

### 4.1. Economic Indicator (EI)

The results for the economic indicator and the related groups are illustrated in Figure 2, which shows that the value of the ME group in the examined cities is lower than IT. This difference is found to be much smaller in comparative cities compared to model cities. This is due to the representativeness of each group. ME refers to the economic quantity of cities, and how much that city and its country move towards the world economy. IT represents the economic quality of the examined cities and the ability to use their knowledge base to optimize their economical production. High investment in IT tends to represent an increase in EM in the long term [6]. Thus, cities with a low ME value should focus their investments on technological and innovative knowledge, aiming to improve their economic performance. Similarly, cities with high macroeconomic levels must also maintain good levels in IT to conserve their positions.

Higher values for ME are obtained from major cities in countries of high attractiveness and international financial movements, such as the five major cities in the world [95]: New York, London, Tokyo, Los Angeles and Paris. This condition is easily observed when comparing the values obtained for Boston, an American city with a strong world economy, and cities in developed countries that have a smaller national economy, such as Stockholm [97]. Nevertheless, Boston is not the main city of its country economically [100], which justifies the low absolute values obtained by all cities (Boston achieves the best performance among cities with a value of (0.656). It is also verified that although the national greatness influences the indicator, the economic importance of the city individually in the global context is what governs it. Rio de Janeiro is just ahead of Cape Town, being the second to last of all analyzed.

When analyzing the values for IT, as presented in Figure 2, the great performance of Stockholm is highly realized, with a value of (0.855). This comes back to the fact that this city is among the most

innovative and technological cities in the world [101,102]. The Swedish capital seeks to improve its economy and urban competitiveness by investing in new patents and research, presenting elements that make up a smart city [103]. In this group, the discrepancy between the model and comparative cities is more evident, presenting averages of (0.658) and (0.295), respectively. This scenario is discouraging for comparative cities and Rio de Janeiro, as they have low values in both groups in this category. Such cities should intensify their investments in innovation, technology, research, and development in order to attract more foreign investments and retain human capital, as well as contributing to greater economic production that guarantees more financial inputs to be invested in other areas of urban development.

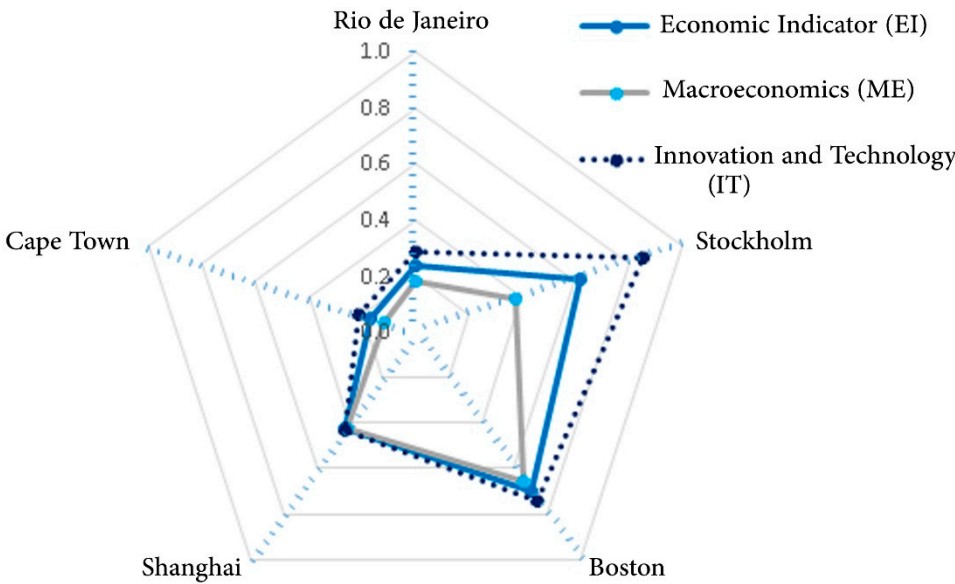

**Figure 2.** Comparative radar chart for the economic indicator (EI).

At the composite category, Boston and Stockholm illustrate the best results, while Rio de Janeiro outperforms only Cape Town. The main strengths of the city are due to national indicators, driven by the Brazilian position in relation to other emerging economies such as South Africa [98] and the size of the city and its population [81]. The city of Rio de Janeiro has intermediate values in foreign investment, direct research and development, and worldwide enterprises. Furthermore, Rio de Janeiro should pursue a policy similar to Stockholm city by increasing investment in research and development, patents and fostering entrepreneurship [104], in order to attract a wide range of foreign investment and headquarters of major world companies. This could, in turn, guarantee better GPD values, increasing the urban competitiveness of the city itself.

### 4.2. Social Indicator (SI)

The results obtained for the social indicator and its groups for cities can be seen in Figure 3. The two groups forming the category have discrepant values, with HSC having values less than DI for all cities. The Brazilian fragility in promoting a good quality of life and social development to its inhabitants is marked when analyzing the values obtained for HSC, presenting the worst results among the cities (0.216). Rio de Janeiro has poor levels of health [73,105,106], allied to a weak distribution of connectivity and broadband access to its population that justifies this position [74,107].

Education and university indicators are inversely related, as presented in Figure 3. For instance, cities with higher values for universities already have a well-defined and solid educational base, needing smaller investments in education to improve their education, such as Boston and Shanghai; for these cities, efforts are instead focused on maintaining their high levels. Stockholm stands out in its quest to increase its performance by gaining greater international recognition for its universities,

maintaining an investment of 7.70% of national GDP in education [71]. Unfortunately, Rio de Janeiro needs a great effort, since it has the worst performance in the universities indicator, with an extremely low score of (0.089), while Brazilian national investment in education (5.90% of GDP) is only slightly above the average of 5.26% of the other examined cities [71]. Additionally, health incentives were undermined by the crisis in the city of Rio de Janeiro, taking into consideration that, in 2016, the city had the worst percentage investment in the country [108]. Considering the ID group, the value for the Brazilian city confirms the solid democracy that Brazil presents in relation to other emerging countries [109]. Moreover, the main advantage of the country is the low socioeconomic dependence experienced in comparison with the more developed countries, especially in Europe, which are undergoing an aging process of their population [110]. However, the country has a young and economically active population and a high percentage of unemployment levels [75], which could influence the use of the productive force in their cities.

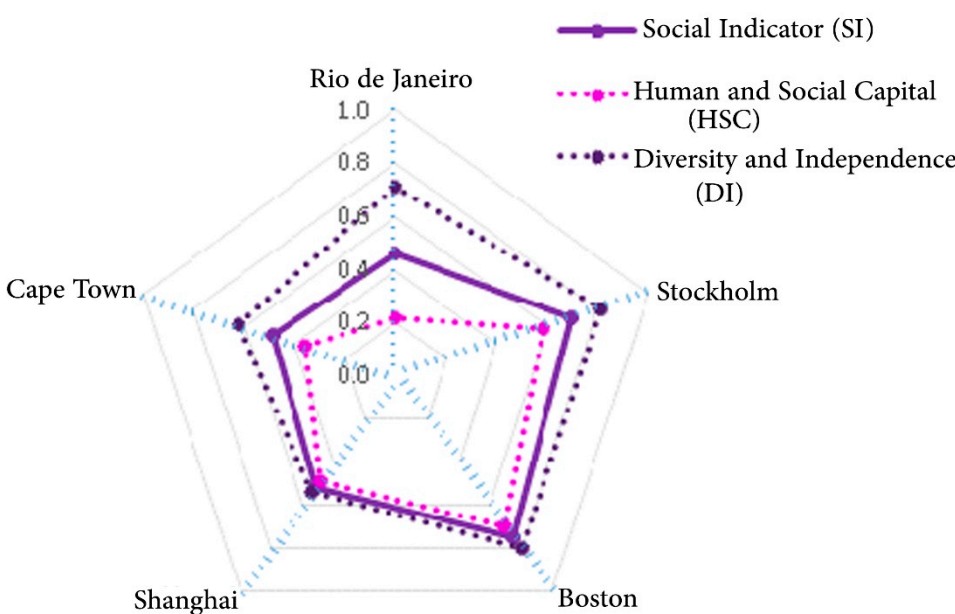

**Figure 3.** Comparative radar chart for the social indicator (SI).

Comparing the Brazilian city with Shanghai, one can see that the reality of Brazil and China are quite different. The Chinese city has high values in HSC and low values in DI, while the situation is totally the opposite in Brazil. This is due to the stricter character of the Chinese government, which, although more interventionist and less liberal [109], promotes good levels of education, health, and connectivity to its population. On the other side, the Brazilian government guarantees a broader democracy, with great freedom and acceptance of immigrants, but with low levels of human and social development. For the composite category, Boston leads the group, whereas Rio de Janeiro is the worst performer, followed by Cape Town and Shanghai. The lessons learned from this category interfere with the directions to be taken in Brazil, where greater investment should be made in health and education, promoting improvements in the public health network and enabling better performance of national universities, technically empowering the population. Allied to this should come incentives to industry, commerce and service, taking advantage of the country's large production mass in order to leverage its economy and better remunerate its population. Democratic foundations must be maintained, with permissive immigration policies.

### 4.3. Environmental Indicator (EnI)

The results obtained for the environmental indicator and its groups for cities can be seen in Figure 4, which illustrates that the value of the SUD group in all cities is higher than the QLP group.

Analyzing the values of the indicators that form the SUD group shows that there is a reversal of positions related to the most developed and emerging countries for the environmental impact indicator. The model cities, being in developed countries, absorb the impacts generated by the large industry of their countries, presenting much higher greenhouse gas emission levels compared to the values of the comparative cities. The only exception is Stockholm, which has a modern, clean industry with few environmental impacts [111]. The city of Rio de Janeiro appears as a leader in this singular indicator. This comes back to the fact that the clean hydropower-based energy matrix in Brazil [112] and policies to reduce greenhouse gas emissions [113] have boosted the performance of the Brazilian cities. Despite the good values for environmental impact, Brazilian cities fail to perform well in the sustainability indicator. On the other hand, the model cities are able to balance their environmental impacts by investing in smart technologies and sustainable urban elements.

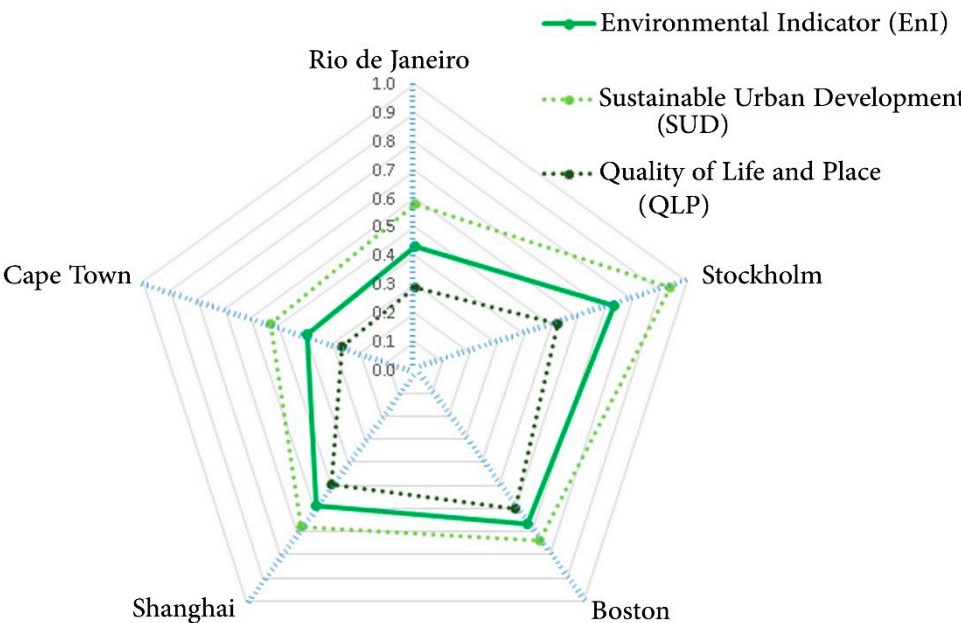

**Figure 4.** Comparative radar chart for the environmental indicator (EnI).

For the group indicators for density and urban mobility, Boston stands out more than the others. While Rio de Janeiro and Cape Town are generally very populous cities [81] and with poorly developed urban mobility infrastructures, with no intermodality-promoting elements and few available mobility options, Boston provides its population with an urban fabric, connected with various modes of travel and intelligent systems [114], and a more balanced and sustainable population density [81]. The damage caused by a heavily populated city is intensified when its mobility is not developed. The high demand for locomotion and the scarce and inefficient supply cause an increase in urban traffic, crowding of wagons and public vehicles and increased time spent on transportation.

Rio de Janeiro should be based on the Asian urban mobility model, exemplified by the analysis of the city of Shanghai, which, although it has a very large population, its inhabitants are able to move efficiently daily, with diverse transport modalities and a well-developed infrastructure [115]. Investing in intermodal connections to Rio to serve all regions without overburdening one type of transport, as well as expanding metro and rail networks and upgrading their components is essential for improving their urban mobility. In addition to this, it should incorporate smart urban elements into its infrastructure to increase the level of sustainability in the city, such as the automation and use of LED lamps in street lighting [116], a more efficient urban waste collection and sorting service [117], and more stormwater reuse and disposal devices [118].

For the QLP group, the starting point for the analysis is the cost of living and housing indicators. A city with a high cost of goods, services, and housing must, in return, have a high quality of life and

security, generating a return for the population, as is the case with Scandinavian cities [96]. However, this dynamic is easily observed only in less densely populated cities. However, cities with a large population, such as comparative cities [78], generate a high demand for housing and services, thus increasing the costs of the city as a whole without bringing qualitative returns to the population. High costs could lead to cluttered land occupation and inappropriate site construction, which in some cases leads to slum processes. Rio de Janeiro must tackle its security problems as a matter of priority. The city is considered one of the most dangerous in the world [119] and its quality of life is largely influenced by this old problem, confirmed by its worst performance in the indicator, with a value of (0.101). Cerqueira (2018) [120] highlights some solutions that can be adopted in the city to make it safer. The researcher suggests changes to be made to the police, making it smarter and information-driven, such as the creation of an interconnected police system, execution based on strategic planning, and improved counterintelligence and homicide investigation services.

### 4.4. Government Indicator (GI)

The results obtained for the government indicator and its groups for cities are illustrated in Figure 5, which presents the category of government as the most balanced in relation to the discrepancy of the groups that form it. GP and LS have close values that follow the same linearity for all cities. At this level of the analysis, the GP group analyzes the effectiveness and modernization of the national governments of the cities analyzed, as well as urban planning, resilience level (the ability of a city to adapt and remain whole after disasters and crises) and their attractiveness. It is a group of macro indicators, which aim to illustrate general and comprehensive characteristics of the policies adopted by city governments.

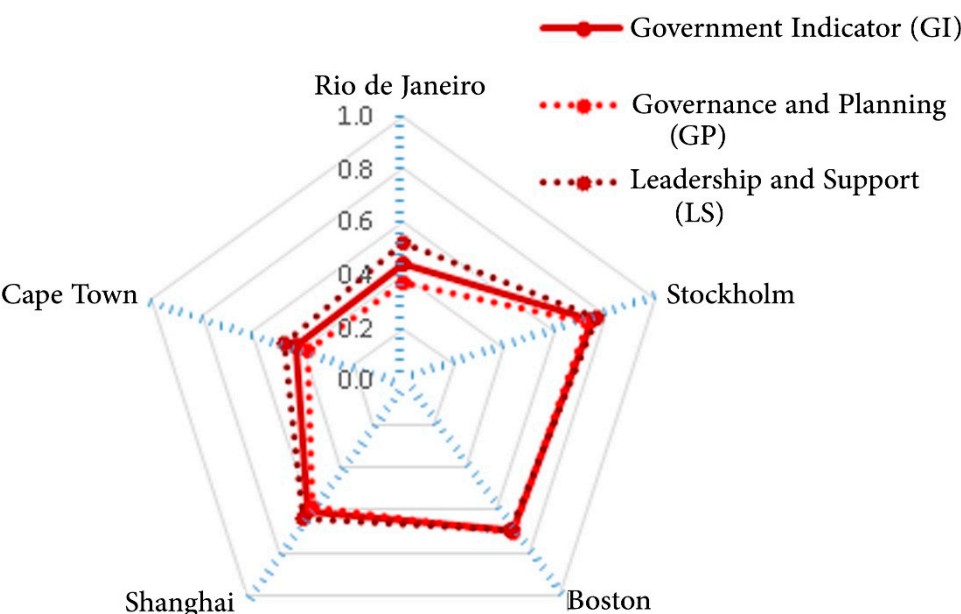

**Figure 5.** Comparative radar chart for the government indicator (GI).

LS corresponds to specific elements of government monitoring, evaluating their transparency and the control mechanisms of the economy and income distribution to the population. It is worth noting the evaluated dynamics for the corruption and tax indicators; high values in both indicators show a very unfavorable situation, with the population paying high rates to the government. Such a situation is observed for Brazilian cities only, which have negative performances in both. In addition, the category of government has the characteristic of influencing the others. The government part of the city is responsible for urban planning, public investment guidelines, and city management and maintenance. Hence, Brazilian cities must prioritize the development of a more transparent

government. High taxes should guarantee them the inputs needed for high public investments in infrastructure and urban development. On the other side, the high value does not necessarily reflect a bad feature, as seen in Stockholm; a city with high taxes that produce high investments and returns. However, Brazilian inflation is improving [119], although the country continues to have a high level of social inequality [120] and should maintain policies of assistance and integration among the different social levels of the population.

*4.5. Urban Development Index (UDI)*

The values of the four overlapping categories in the same graph can be seen in Figure 6, which illustrates the four categories grouped together. It is possible to see the underperformance of the economic category for all cities, especially comparative cities, and the great disparity between the two sets of cities. For the other categories, the values are better distributed and generally follow the same linearity, although they vary between each city.

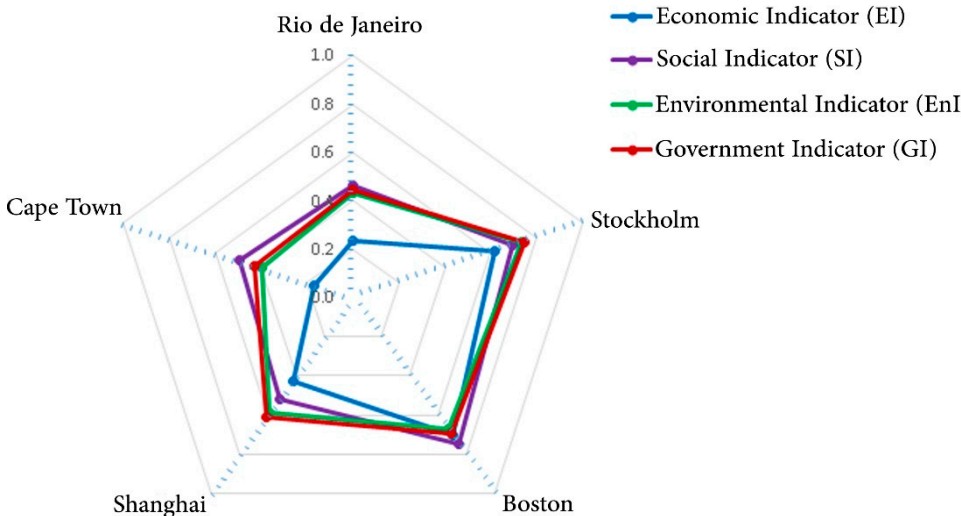

**Figure 6.** Comparative radar chart with overlapping categories.

At this level of the analysis, it is worth highlighting the results for the GI obtained by Shanghai, which has a more interventionist and less democratic government among the four examined cities [109]. Rio de Janeiro has the worst results for the economic category, similar to other cities (except Boston), while its other three categories have very close values. The city should emphasize the development of the parameters that make up the ME, HSC, and GP groups. The results obtained for the UDI of each of the five cities can be seen in Figure 7.

At the final composite index, the sovereignty of the model cities over the comparative ones is clear, with Shanghai intermediate between the two sets. There is also a non-confirmation of the relationship between the UDI and the HDI, except for Cape Town, which was the worst ranking for both indicators. The highest values for the UDI were obtained by Boston and Stockholm, all with values close to (0.700). It is important to note that the two cities have different characteristics, with different leading categories. Boston has better values for SI, while Stockholm has better values for GI. However, the city of Rio de Janeiro occupies, at the end of the modeling, the penultimate position in the UDI ranking, just ahead of Cape Town. This result confirms the precariousness of the city compared to the other global cities studied and legitimizes the emergence of a more efficient urban development. The worst indicators obtained for Rio de Janeiro were universities (0.089), safety (0.101) and health (0.123). Therefore, planning and investment should be intensified in order to address these deficiencies.

Regarding the security indicator, the result was already expected, as the city has one of the worst notorieties regarding insecurity and organized crime, which is subject to its daily population. The recent initiatives include the installation of Pacifying Police Units (PPUs), a public action to combat

and control drug trafficking in the communities of Rio de Janeiro, and the monitoring done by SRC (Special Resources Coordination), a special police unit that should, in the long term, improve the local situation in Rio de Janeiro. However, there is a critical need to upgrade the police fleet and invest more in empowering and making it smarter by creating more accurate tactical police components. Finally, public investments in health should be increased. The Brazilian Unified Health System was a pioneer worldwide in its creation. Nevertheless, it is broken and unable to serve the population properly. Public hospitals are in a precarious situation, lacking qualified drugs, equipment, and facilities. A greater effort is needed from the municipal, state and national agencies to carry out their maintenance, intensifying investments for health.

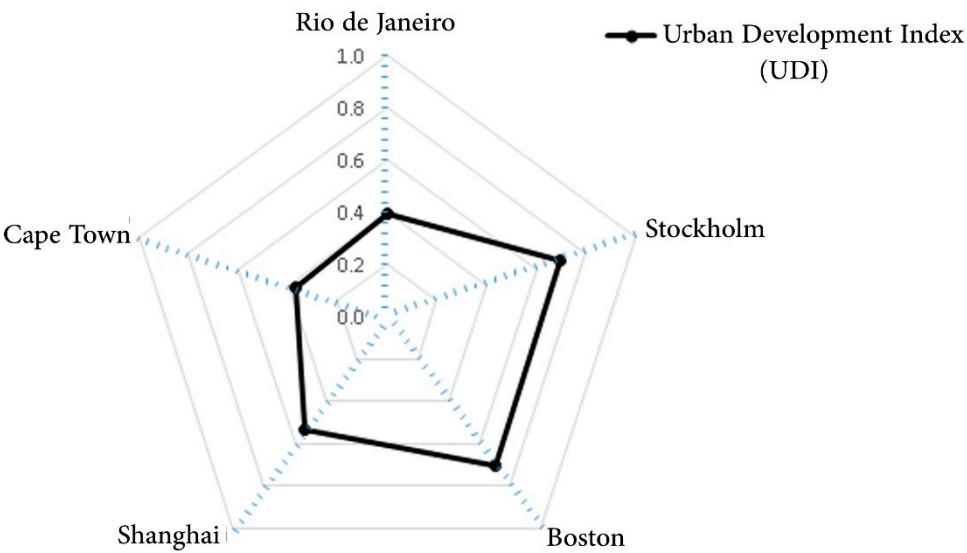

**Figure 7.** Comparative radar chart for the Urban Development Index (UDI).

*4.6. Implications for Future Applications*

The modeling and analysis of the results allowed us to expose the main deficiencies of the studied cities, especially the reference city, Rio de Janeiro. The advance from these discussions would be to promote solutions to improve their urban performance, observing the most successful cases among the other examined cities and absorbing practices that would be applicable to the city in question. However, such practices must be brought into the context of the reference city and made possible; after all, the urban success achieved through the best practices of another city in its local context needs adjustments for its implementation to the reality of Rio de Janeiro.

**5. Conclusions**

In this work, the level of urban performance offered by the city of Rio de Janeiro compared to other global cities was examined. An index modeling the urban performance of cities, as derived from data obtained from existing bibliography as proposed. To achieve the index modelling, a knowledge-based urban development (KBUD) methodology was adopted in order to obtain indicators of urban development for each examined city (Rio de Janeiro, Stockholm, Shanghai, Boston, and Cape Town). This work models cities with a high urban performance and comparative cities with an urban performance similar to Rio de Janeiro. In addition, a new analysis model was designed to suit the proposed objectives and selected cities. From the selected cities, this work presented a method for measuring their development, focusing on Rio de Janeiro. Thus, it was possible to obtain a reference for how the city of Rio de Janeiro is positioned in relation to the others. The core of the research is precisely the information taken from the modeling and its interpretation in order to illustrate the urban behavior of Rio de Janeiro. For the composition of the work, the following executive sequence was:

(i) bibliographic review about the urban situation in Rio de Janeiro and about KBUD; (ii) determination of the modeling structure and selection of indicators to be implemented; (iii) selection of cities to be used as benchmarking; (iv) data collection for model feeding; (v) data normalization; (vi) calculation of indicator groups, categories, and composite index; (vii) interpretation of results and discussions.

Regarding the Urban Development Index (UDI), the city of Rio de Janeiro was ranked second last in relation to the five cities studied, confirming the need to intensify planning and investments in order to combat the deficiencies indicated by the indicators. This result confirms the precariousness of the city compared to the other global cities studied and legitimizes the emergence of more efficient urban development. The city of Rio de Janeiro should promote investments in research and development. Investments in health and education should also be prioritized, but there should also be incentives for industry, trade and service. Finally, Rio de Janeiro must tackle security problems as a matter of priority.

The major challenge of the study in question was data collection, especially considering the scarcity of comparable information applicable to the study. The need was identified to previously evaluate the parameters to be collected, verifying if the date for all analyzed cities originated from the same sources. This consideration had a major influence on the selection of the indicators. Thus, estimates or uncertain information were eliminated. On the other hand, the choice of indicators was limited to include all the cities studied. In addition, some references used to obtain the secondary data utilized are slightly outdated. Although they do not strongly modify the results obtained, it should be taken into account that some data could be slightly obsolete and may have changed from when it was taken to today. In any case, the oldest source used is from 2013, which for general purposes is an acceptable time range for the analysis.

A natural evolution for the continuity of the research would be to include other cities in the analysis, expanding the cases of benchmarking and allowing for the adoption of another city as a focus. The methodology and model implemented are also subject to improvements. For example, one can elaborate other models that encompass different parameters and indicators, with some specific focus on any of the four mid-level categories presented. In addition, changing the normalization can be done by using standard deviation statistical normalization. The influence of the indicators on the composition of groups and categories can be changed by applying different weights to the values in order to better adjust the importance of each indicator for the analysis.

**Author Contributions:** Conceptualization, R.M. and E.V.; Data curation, M.K.N.; Formal analysis, R.M. and A.W.A.H.; Investigation, R.M., A.H. and E.V.; Methodology, R.M. and E.V.; Resources, R.M.; Software, M.K.N.; Supervision, A.H. and E.V.; Validation, A.W.A.H. and E.V.; Writing—original draft, R.M. and M.K.N.; Writing—review & editing, A.W.A.H., A.H. and E.V. All authors have read and agreed to the published version of the manuscript.

**Funding:** This research was funded by COPPETEC Research Project number (17902) Also the financial support from CNE FAPERJ 2019-E-26/202.568/2019 (245653) Fundação de Amparo à Pesquisa do Estado do Rio de Janeiro, and CNPq (Brazilian National Council for Scientific and Technological Development) grant number [307084/2015-9].

**Acknowledgments:** The authors want to thank Departamento de Construção Civil, Poli-UFRJ, in facilitating equipment installations and resources for the development of this project.

**Conflicts of Interest:** The authors declare no conflicts of interest.

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
