# Peer review of "Urban Development Index (UDI): A Comparison between the City of Rio de Janeiro and Four Other Global Cities"

_sustainability, doi:10.3390/su12030823_

Round 1
Reviewer 1 Report
In my opinion, this work presents importance and significance in the field of urban development inde. However, the contents of this manuscript should be further enhanced. For instance, the abstract should clearly state, by 2-3 sentences, the essence of the problem you are addressing, what you did and what you found and recommend. In the Introduction part, the authors should provide a critical overview on challenges, barriers, opportunities and recent knowledge breakthrough in this specific research field. Also, a proof reading by a native English speaker should be conducted to improve both language and organization quality. Several additional comments were provided for further improvement as follows:
The abstract should clearly state the essence of the problem you are addressing, what you did and what you found and recommend. That will help a prospective reader of the abstract to decide if they wish to read the entire article. Please check the use of abbreviations throughout the manuscript. The provided results (e.g., Table 1, etc.) should be further illustrated and/or elaborated. Please provide a new Section, briefly indicating the implications of the provided results to the future applications. Please provide the Conclusion in only one paragraph with more quantitative information. The current Conclusion is too lengthy. A proof reading by a native English speaker should be conducted to improve both language and organization quality.Author Response
Responses to the comments from the reviewers on the Review Form:
First Revisor
In my opinion, this work presents importance and significance in the field of urban development index.
Response: Thanks for your time in reviewing our manuscript.
However, the contents of this manuscript should be further enhanced. For instance, the abstract should clearly state, by 2-3 sentences, the essence of the problem you are addressing, what you did and what you found and recommend.
Response: Thank you for this comment. The abstract has been rewritten emphasizing work motivation, and addressing the comments as follows:
The motivation of this work is the fact that the city of Rio de Janeiro, although widely known and admired around the world for its natural beauty, has a wide negative notoriety regarding its urban functionality. There is a critical need for investment in the city's infrastructure in order to improve the quality of life of its population.
And
In the proposed methodology for modeling the Urban Development Index (UDI), this work presents the model structure made from a knowledge-based urban development assessment model (KBUD / AM), the determination of the indicators, the selection of the cities, the data collection from primary and secondary sources and the use of statistical techniques for data normalization and scaling. The research aims to quantify, compare and evaluate the level of urban development of Rio de Janeiro, performing benchmarking with other four global cities (Stockholm, Shanghai, Boston, and Cape Town).
And
Based on the result, Rio de Janeiro, ranks as the second to last among the five cities studied, with an UDI of 0.395, only slightly better than Cape Town. Results confirm that the city of Rio de Janeiro has several deficiencies, especially in the education, safety and health sectors and is very far from most of the other developed cities. The city of Rio de Janeiro should promote investments in research and development. Finally, this work confirms that Rio de Janeiro must tackle security problems as a matter of priority.
In the Introduction part, the authors should provide a critical overview on challenges, barriers, opportunities and recent knowledge breakthrough in this specific research field.
Response: Thank you for this comment. An additional subsection was added to the introduction as follows:
1.1. Justification for the study
Quality of life in cities is a multidimensional subject and, in this respect, its evaluation is a challenge [26]. In the recent literature, a large number of studies focused on the quality of life in cities [27, 28]. The environmental index is strongly connected with quality of life [26,29,30]. Other indices also affect the quality of life in cities such as governance, economy, mobility, safety, quality and service provision [24]. Good urban performance is directly related to effective and balanced joint development between the social, environmental and economic sectors of sustainable development [31-34].
Studies in the literature have highlighted that many factors can directly or indirectly affect sustainable urban development levels, indicating that the potential index system scale is large [35]. This research aims to quantify, compare and evaluate the level of urban development of Rio de Janeiro in relation to four other chosen global cities (Stockholm, Shanghai, Boston, e Cape Town). The novelty of this work is determining an index, denominated as UDI, that quantifies the urban functionality of the city, and that represents urban development. The research focuses on optimizing the investments in the infrastructure and increasing the urban performance offered by the city of Rio de Janeiro. Data-based modeling was applied from the knowledge-based urban development (KBUD) methodology, in order to obtain a UDI for Rio and the other selected cities. As a scientific contribution, this study will make it possible to observe the particularities of the urban functioning of Rio de Janeiro compared to the other cities studied, as well as identifying the possibilities of urban evolution.
Also, a proof reading by a native English speaker should be conducted to improve both language and organization quality.
Response: Thanks for this comment. We have conducted a proof reading by a native English speaker.
Several additional comments were provided for further improvement as follows:
The abstract should clearly state the essence of the problem you are addressing, what you did and what you found and recommend. That will help a prospective reader of the abstract to decide if they wish to read the entire article.
Response: Thank you for this comment. The abstract has been edited as follows:
Abstract: One of the methods to assess the urban development of a city is to allocate indicators that quantify its efficiency to perform various functions such as urban mobility, security, sustainability, economy, among others. The motivation of this work is the fact that the city of Rio de Janeiro, although widely known and admired around the world for its natural beauty, has a wide negative notoriety regarding its urban functionality. There is a critical need for investment in the city's infrastructure in order to improve the quality of life of its population. The novelty of this work is in proposing an index that quantifies the urban functionality of the city of Rio de Janeiro and that represents urban development. The research focuses on optimizing investments in infrastructure and hence increasing the urban performance offered by the city of Rio de Janeiro. In the proposed methodology for modeling the Urban Development Index (UDI), this work presents the model structure made from a knowledge-based urban development assessment model (KBUD / AM), the determination of the indicators, the selection of the cities, the data collection from primary and secondary sources and the use of statistical techniques for data normalization and scaling. The research aims to quantify, compare and evaluate the level of urban development of Rio de Janeiro, performing benchmarking with other four global cities (Stockholm, Shanghai, Boston, and Cape Town). Cities are ranked for their UDI to make the comparison more straightforward. Based on the result, Rio de Janeiro, ranks as the second to last among the five cities studied, with an UDI of 0.395, only slightly better than Cape Town. Results confirm that the city of Rio de Janeiro has several deficiencies, especially in the education, safety and health sectors and is very far from most of the other developed cities. The city of Rio de Janeiro should promote investments in research and development. Finally, this work confirms that Rio de Janeiro must tackle security problems as a matter of priority.
Please check the use of abbreviations throughout the manuscript.
Response: Thanks for this comment. We have checked the abbreviations in the revised manuscript.
The provided results (e.g., Table 1, etc.) should be further illustrated and/or elaborated.
Response: Thank you for this important comment. A description and additional table are provided, in subsection 2.4., to further clarify the relationship between categories, groups and unique indicators as follows:
The next step after selecting cities is to collect data from the 32 indicators using primary and secondary sources. At this level of the analysis, Table 1 presents the eight indicator groups. in addition, Table 1 illustrates the four main categories of analysis that provide independent indicators such as (EI), (SI), (EnI) and (GI), as well as their 32 unique indicators. Table 2 provides an explanatory description of each of the 32 singular indicators and bibliographic references.
Table 1. Description of the eight indicator groups, four analysis categories, and their 32 singular indicators
|
Analysis Categories |
Indicator Groups |
Singular Indicators |
|
Economic Indicator (EI) |
Macroeconomics (ME) |
GDP |
|
International companies |
||
|
Foreign direct investment |
||
|
Urban competitiveness |
||
|
Innovation and Technology (IT) |
Innovation |
|
|
Research and Development |
||
|
Smart city |
||
|
Patents |
||
|
Social Indicator (SI) |
Human and Social Capital (HSC) |
Education |
|
Universities |
||
|
Health |
||
|
Connectivity |
||
|
Diversity and Independence (DI) |
Immigration |
|
|
Freedom |
||
|
Socioeconomic Dependence |
||
|
Unemployment |
||
|
Environmental Indicator (EnI) |
Sustainable Urban Development (SUD) |
Environmental impact |
|
Sustainability |
||
|
Urban density |
||
|
Urban mobility |
||
|
Quality of Life and Place (QLP) |
Quality of life |
|
|
Cost of living |
||
|
Residency |
||
|
Safety |
||
|
Government Indicator (GI) |
Governance and Planning (GP) |
Government Effectiveness |
|
Electronic Governance |
||
|
Tourism |
||
|
Urban planning and resilience |
||
|
Leadership and Support (LS) |
Corruption |
|
|
Taxes |
||
|
Inflation |
||
|
Social equality |
Please provide a new Section, briefly indicating the implications of the provided results to the future applications.
Response: Thank you for this important comment. An additional item, subsection 2.6., has been inserted indicating how results can lead to future work as follows:
4.6. Implications for Future Applications
The modeling and analysis of the results allowed to expose the main deficiencies of the studied cities, especially the reference city, Rio de Janeiro. The advance from these discussions would be to promote solutions to improve their urban performance, observing the most successful cases among the other examined cities and absorbing practices that would be applicable to the city in question. However, such practices must be brought into the context of the reference city and made possible, after all, the urban success achieved through the best practices of another city in its local context needs adjustments for its implementation to the reality of Rio de Janeiro.
Please provide the Conclusion in only one paragraph with more quantitative information. The current Conclusion is too lengthy.
Response: Thank you for this comment. The conclusion was reduced and qualitative results were presented as follows:
In this work, the level of urban performance offered by the city of Rio de Janeiro compared to other global cities was examined. An index modeling the urban performance of cities, as derived from data obtained from existing bibliography as proposed. To achieve the index modelling, a knowledge-based urban development (KBUD) methodology was adopted in order to obtain indicators of urban development for each examined city (Rio de Janeiro, Stockholm, Shanghai, Boston, and Cape Town). This work models a city with high urban performance and comparative cities with urban performance similar to Rio de Janeiro. In addition, a new analysis model was designed to suit the proposed objectives and selected cities. From the selected cities, this work presented a method for measuring their development, focusing on Rio de Janeiro. Thus, it was possible to obtain a reference for how the city of Rio de Janeiro is positioned in relation to the others. The core of the research is precisely the information taken from the modeling and its interpretation in order to illustrate the urban behavior of Rio de Janeiro. For the composition of the work, the following executive sequence was: (i) bibliographic review about the urban situation in Rio de Janeiro and about KBUD; (ii) determination of the modeling structure and selection of indicators to be implemented; (iii) selection of cities to be used as benchmarking; (iv) data collection for model feeding; (v) data normalization; (vi) calculation of indicators groups, categories, and composite index; (vii) interpretation of results and discussions.
Regarding the Urban Development Index (UDI), the city of Rio de Janeiro was ranked second last in relation to the five cities studied, confirming the need to intensify planning and investments in order to combat the deficiencies indicated by the indicators. This result confirms the precariousness of the city compared to the other global cities studied and legitimizes the emergence for a more efficient urban development. The city of Rio de Janeiro should promote investments in research and development. Investments in health and education should also be prioritized, but there should also be incentives for industry, trade and service. Finally, Rio de Janeiro must tackle security problems as a matter of priority.
The major challenge of the study in question was data collection, especially considering the scarcity of comparable information applicable to the study. It was identified the need to previously evaluate the parameters to be collected, verifying if the date for all analyzed cities originated from the same sources. This consideration had a major influence on the selection of indicators. Thus, estimates or information with uncertainty were eliminated. On the other hand, the choice of indicators was limited to include all the cities studied. In addition, some references used to obtain the secondary data utilized are slightly outdated. Although they do not strongly modify the results obtained, it should be taken into account that some date could be slightly obsolete and may have changed from today. In any case, the oldest source used is 2013, which for general purposes is an acceptable time range for the analysis.
A natural evolution for the continuity of the research would be to include other cities in the analysis, expanding the cases of benchmarking and allowing the adoption of another city as a focus. The methodology and model implemented are also subject to improvements. For example, one can elaborate other models that encompass different parameters and indicators, with some specific focus on any of the four mid-level categories presented. In addition, changing the normalization can be done by using standard deviation statistical normalization. The influence of the indicators on the composition of groups and categories can be changed by applying different weights to the values in order to better adjust the importance of each indicator for the analysis.
A proof reading by a native English speaker should be conducted to improve both language and organization quality.
Response: Thanks for this comment. We have conducted a proof reading by a native English speaker. Please find the revised manuscript now.
Reviewer 2 Report
I found this paper interesting and well written. It touches an all times classic subject that of urban performance and ranking of cities. Some minor comments to be taken into consideration by the authors.
Re-write the scope of the paper and introduce research questions – at the end place your achievements versus the research questions. The research gap is not entirely obvious. I kind of missed a section between the introduction and the model development which will form the background to the study. I understand that there may be a words’ limitation however some literature review should be included (not in particular for the model development but with reference to quality of life etc.). Useful references to be includedGiannias, D.A. and Sfakianaki, E. (2013), “Regional and environmental classifications of the 27 EU countries”, The Journal of Developing Areas, Vol. 47, No. 2, pp. 139-157.
Giannias, D.A. and Sfakianaki, E. (2014), “Classifications of environmental quality effects: the case of Canadian cities”, Ε&Μ Economics and Management Journal, Vol. XVII, No. 2, pp. 45-60.
Sylla, M., Lasota, T. and Szewranski, S. (2019), “Valuing Environmental Amenities in Peri-Urban Areas: Evidence from Poland”, Sustainability 2019, 11(3).
With this extra literature introduction there will be a better balance between the theoretical background and the research undertaken. The former is, to my opinion short, whereas there should be a balance between the two.
At the concluding section although conclusions and limitations have been adequately presented, there are no proposals for future research which should be included also. English is generally ok although a good proof reading will certainly help.Author Response
Second Revisor
I found this paper interesting and well written. It touches an all times classic subject that of urban performance and ranking of cities. Some minor comments to be taken into consideration by the authors.
Response: Thanks for your time in reviewing our manuscript.
Re-write the scope of the paper and introduce research questions – at the end place your achievements versus the research questions. The research gap is not entirely obvious.
Response: Thanks for this comment. The abstract was rewritten emphasizing the problem situation and also emphasizes the results obtained with the research as follows:
Abstract: One of the methods to assess the urban development of a city is to allocate indicators that quantify its efficiency to perform various functions such as urban mobility, security, sustainability, economy, among others. The motivation of this work is the fact that the city of Rio de Janeiro, although widely known and admired around the world for its natural beauty, has wide negative notoriety regarding its urban functionality. There is a critical need for investment in the city's infrastructure in order to improve the quality of life of its population. The novelty of this work is in proposing an index that quantifies the urban functionality of the city of Rio de Janeiro and that represents urban development. The research focuses on optimizing investments in infrastructure and hence increasing the urban performance offered by the city of Rio de Janeiro. In the proposed methodology for modeling the Urban Development Index (UDI), this work presents the model structure made from a knowledge-based urban development assessment model (KBUD / AM), the determination of the indicators, the selection of the cities, the data collection from primary and secondary sources and the use of statistical techniques for data normalization and scaling. The research aims to quantify, compare and evaluate the level of urban development of Rio de Janeiro, performing benchmarking with other four global cities (Stockholm, Shanghai, Boston, and Cape Town). Cities are ranked for their UDI to make the comparison more straightforward. Based on the result, Rio de Janeiro ranks as the second to last among the five cities studied, with a UDI of 0.395, only slightly better than Cape Town. Results confirm that the city of Rio de Janeiro has several deficiencies, especially in the education, safety and health sectors and is very far from most of the other developed cities. The city of Rio de Janeiro should promote investments in research and development. Finally, this work confirms that Rio de Janeiro must tackle security problems as a matter of priority.
I kind of missed a section between the introduction and the model development which will form the background to the study. I understand that there may be a words’ limitation however some literature review should be included (not in particular for the model development but with reference to the quality of life etc.). Useful references to be included
Giannias, D.A. and Sfakianaki, E. (2013), “Regional and environmental classifications of the 27 EU countries”, The Journal of Developing Areas, Vol. 47, No. 2, pp. 139-157.
Giannias, D.A. and Sfakianaki, E. (2014), “Classifications of environmental quality effects: the case of Canadian cities”, Ε&Μ Economics and Management Journal, Vol. XVII, No. 2, pp. 45-60.
Sylla, M., Lasota, T. and Szewranski, S. (2019), “Valuing Environmental Amenities in Peri-Urban Areas: Evidence from Poland”, Sustainability 2019, 11(3).
With this extra literature introduction, there will be a better balance between the theoretical background and the research undertaken. The former is, to my opinion short, whereas there should be a balance between the two.
Response: Thank you for this important comment. We have added a subsection in the introduction using the suggested references in order to make a better balance between the theoretical background and the research undertaken as follows:
1.1. Justification for the study
Quality of life in cities is a multidimensional subject and, in this respect, its evaluation is a challenge [26]. In the recent literature, a large number of studies focused on the quality of life in cities [27, 28]. The environmental index is strongly connected with the quality of life [26,29,30]. Other indices also affect the quality of life in cities such as governance, economy, mobility, safety, quality and service provision [24]. Good urban performance is directly related to effective and balanced joint development between the social, environmental and economic sectors of sustainable development [31-34].
Studies in the literature have highlighted that many factors can directly or indirectly affect sustainable urban development levels, indicating that the potential index system scale is large [35]. This research aims to quantify, compare and evaluate the level of urban development of Rio de Janeiro in relation to four other chosen global cities (Stockholm, Shanghai, Boston, e Cape Town). The novelty of this work is determining an index, denominated as UDI, that quantifies the urban functionality of the city, and that represents urban development. The research focuses on optimizing the investments in the infrastructure and increasing the urban performance offered by the city of Rio de Janeiro. Data-based modeling was applied from the knowledge-based urban development (KBUD) methodology, in order to obtain a UDI for Rio and the other selected cities. As a scientific contribution, this study will make it possible to observe the particularities of the urban functioning of Rio de Janeiro compared to the other cities studied, as well as identifying the possibilities of urban evolution.
In the concluding section, although conclusions and limitations have been adequately presented, there are no proposals for future research which should be included also.
Response: Thank you for this comment. We have added an extra subsection considering the proposed work developments as follows:
4.6. Implications for Future Applications
The modeling and analysis of the results allowed us to expose the main deficiencies of the studied cities, especially the reference city, Rio de Janeiro. The advance from these discussions would be to promote solutions to improve their urban performance, observing the most successful cases among the other examined cities and absorbing practices that would be applicable to the city in question. However, such practices must be brought into the context of the reference city and made possible, after all, the urban success achieved through the best practices of another city in its local context needs adjustments for its implementation to the reality of Rio de Janeiro.
The conclusion has been reduced and summarized as follows:
In this work, the level of urban performance offered by the city of Rio de Janeiro compared to other global cities was examined. An index modeling the urban performance of cities, as derived from data obtained from the existing bibliography as proposed. To achieve the index modeling, a knowledge-based urban development (KBUD) methodology was adopted in order to obtain indicators of urban development for each examined city (Rio de Janeiro, Stockholm, Shanghai, Boston, and Cape Town). This working model a city with high urban performance and comparative cities with urban performance similar to Rio de Janeiro. In addition, a new analysis model was designed to suit the proposed objectives and selected cities. From the selected cities, this work presented a method for measuring their development, focusing on Rio de Janeiro. Thus, it was possible to obtain a reference for how the city of Rio de Janeiro is positioned in relation to the others. The core of the research is precisely the information taken from the modeling and its interpretation in order to illustrate the urban behavior of Rio de Janeiro. For the composition of the work, the following executive sequence was: (i) bibliographic review about the urban situation in Rio de Janeiro and about KBUD; (ii) determination of the modeling structure and selection of indicators to be implemented; (iii) selection of cities to be used as benchmarking; (iv) data collection for model feeding; (v) data normalization; (vi) calculation of indicators groups, categories, and composite index; (vii) interpretation of results and discussions.
Regarding the Urban Development Index (UDI), the city of Rio de Janeiro was ranked second last in relation to the five cities studied, confirming the need to intensify planning and investments in order to combat the deficiencies indicated by the indicators. This result confirms the precariousness of the city compared to the other global cities studied and legitimizes the emergence for a more efficient urban development. The city of Rio de Janeiro should promote investments in research and development. Investments in health and education should also be prioritized, but there should also be incentives for industry, trade, and service. Finally, Rio de Janeiro must tackle security problems as a matter of priority.
The major challenge of the study in question was data collection, especially considering the scarcity of comparable information applicable to the study. It was identified the need to previously evaluate the parameters to be collected, verifying if the date for all analyzed cities originated from the same sources. This consideration had a major influence on the selection of indicators. Thus, estimates or information with uncertainty were eliminated. On the other hand, the choice of indicators was limited to include all the cities studied. In addition, some references used to obtain the secondary data utilized are slightly outdated. Although they do not strongly modify the results obtained, it should be taken into account that some date could be slightly obsolete and may have changed from today. In any case, the oldest source used is 2013, which for general purposes is an acceptable time range for the analysis.
A natural evolution for the continuity of the research would be to include other cities in the analysis, expanding the cases of benchmarking and allowing the adoption of another city as a focus. The methodology and model implemented are also subject to improvements. For example, one can elaborate on other models that encompass different parameters and indicators, with some specific focus on any of the four mid-level categories presented. In addition, changing the normalization can be done by using standard deviation statistical normalization. The influence of the indicators on the composition of groups and categories can be changed by applying different weights to the values in order to better adjust the importance of each indicator for the analysis.
English is generally ok although good proofreading will certainly help.
Response: Thanks for this comment. We have conducted proofreading by a native English speaker. Please find the revised manuscript now.
Round 2
Reviewer 1 Report
Since the authors have made significant revision according to the comments raised by reviewers, I am supportive of this manuscript for publication.